# Emotional Bias among Individuals at Risk for Seasonal Affective Disorder—An EEG Study during Remission in Summer

**DOI:** 10.3390/brainsci14010002

**Published:** 2023-12-20

**Authors:** Dagný Theódórsdóttir, Yvonne Höller

**Affiliations:** Faculty of Psychology, University of Akureyri, 600 Akureyri, Iceland

**Keywords:** seasonality, seasonal affective disorder, winter depression, EEG alpha power, emotions

## Abstract

Emotional bias in attention and memory is well researched in depression. Patients with depression prioritize processing of negative information over positive input. While there is evidence that emotional bias exists in seasonal affective disorder (SAD) during winter, it is unclear whether such altered cognition exists also during summer. Moreover, it is unclear whether such bias affects attention, memory, or both. In this study, we investigated 110 individuals in summer, 34 of whom reported suffering from low mood during winter, according to the seasonal pattern assessment questionnaire. While the electroencephalogram was recorded, participants learned 60 emotional pictures and subsequently were asked to recognize them in an old/new task. There were no clear group differences in behavioral measures, and no brain response differences in frontal alpha power during learning. During recognition, at 100–300 ms post stimulus individuals with higher seasonality scores exhibited larger alpha power in response to negative as compared to neutral stimuli, while individuals with low seasonality scores exhibited larger alpha power in response to positive as compared to neutral stimuli. While we cannot draw conclusions whether this is an effect of attention or memory, the finding suggests that early cognitive processes are altered already during summer in individuals with increased likelihood to experience SAD during winter. Our data provide evidence for an all-year-round cognitive vulnerability in this population.

## 1. Introduction

Emotional memory involves cognitive and physiological responses to events that evoke emotions, whether people are conscious about these emotional reactions or not [1]. Experiences involving positive and negative events are known to be more likely to affect memory than mundane events [2]. This mechanism impacts encoding of emotional stimuli, memory consolidation, and retrieval [3]. Measuring emotional memory bias during free recall reveals that performance is likely to be superior for emotional stimuli than neutral content [4,5]. Emotional memory bias can occur as positivity bias, where memory is enhanced for positive stimuli [6], and negativity bias, referring to enhanced memory for negative stimuli [7]. For individuals with affective disorders, emotional memory biases are a potential obstacle to mental health. It is possible that emotional bias is not only a symptom but that it increases vulnerability for the disorder and/or maintains the condition [8,9]. The hypothesized mechanism of action is based on the resource allocation theory [10]. According to this theory, negative memory bias enhances the likelihood of sad mood by activating maladaptive ruminative thinking patterns. These patterns are a core cognitive characteristic of depression. As they occupy cognitive resources, they limit the cognitive capacity available to perform other mental processing [10]. Accordingly, enhanced positivity bias has been documented in acute manic phases of bipolar I disorder [11], and pronounced negativity bias has been shown to occur in individuals with depression [12,13,14].

In addition to memory bias, other cognitive control processes are affected in depression. Memory relies on executive functions such as attention allocation and information monitoring [15]. Attention and executive functions have been reported to be impaired not only during major depressive episodes but also in euthymic patients with a diagnosis of major depression [16]. The body of research on bias in attention and memory among individuals with depression builds on the theoretical framework proposed by Williams et al. [17], which distinguishes attentional from memory processes. The theory suggests that both encoding and retrieval of information rely on automatic as well as conscious processes which are negatively affected in individuals with depression. This theory predicts that attentional biases in encoding are independent from explicit memory (e.g., free recall) but are more related to implicit memory (e.g., recognition). Well in line with this theory, studies in depression revealed a negative attentional bias in depressed participants towards sad faces [18], a decreased ability to identify happy faces [19], and impaired recognition of neutral faces as compared to healthy controls [20]. These results can be interpreted as impaired ability to monitor attention to negative stimuli [21] and reduced reactivity to positive stimuli [22]. While there is evidence that suggests that attentional bias might be an important mediator of the relationship between memory bias and depressive symptoms [23], there are also critical voices calling into question whether there is an attention bias in depression. According to a study employing a meta-analytic approach, memory bias is a direct correlate of subclinical depression, while this is not the case for attention bias [24]. A potential explanation of these conflicting findings is provided by the indirect effect model [25]. According to this model, an emotional bias in attention is related to a bias in interpretative choices, and those lead to biased memory [25].

Attentional bias and implicit memory bias for negative stimuli was observed only in clinically depressed individuals [26], but dysfunctional attentional bias in the form of reduced attention to positive stimuli can be found not only in individuals with depression, but also in dysphoric individuals, and previously depressed patients [27]. In line with the core claims of cognitive models of depression, attentional biases are traits rather than states and should be detectable at any time in affected individuals [27]. It is therefore plausible that identifying and treating these biases should be an important component of treatment and prevention [28]. It was shown that patients with depression benefit from relocating their attention focus by improved subsequent recall [29], a mechanism that could inform cognitive behavioral therapeutic treatments.

Even though emotional cognitive biases have been studied thoroughly in depression, the phenomenon is less understood for seasonal depression in winter [30]. Seasonal depression or seasonal affective disorder (SAD) is a mood disorder characterized by depressive episodes occurring annually at the same time of the year [31]. Most commonly, affected individuals experience depression in winter, which is supposed to be attributable to decreased amounts of sunlight [32]. In addition to sad mood, seasonality symptoms include daytime drowsiness, hypersomnia, anxiety, and irritability [31]. Prior research reported a negativity bias for participants with winter depression, such as diminished recognition of happy faces, amplified recognition memory for negative personality characteristics, and increased vigilance of negative words during winter [33]. In addition, a reduction in free recall performance of positive pictures has been reported during the symptomatic period of SAD [33,34,35]. When interpreting these findings, it is noteworthy that emotional bias in SAD was detected for attention and memory. More specifically, emotional bias was detected during recall and recognition in individuals with SAD. In recognition tasks it was demonstrated that patients with depression are more likely to consider words of negative valence as old, even if the negative word was in fact new [36,37]. Thus, there are mood-congruent false memories towards negative [36] and emotional words [38], but no bias towards neutral words [36,38]. According to this result, individuals with depression create false memories that are negative—a phenomenon that could contribute to their overall vulnerability for the affective disorder. This is possibly also relevant for SAD, but the question is whether these vulnerabilities are seasonal, too, or whether there is an all-year-round bias in attention and/or memory.

Research on emotional cognition bias in SAD has mostly focused on the symptomatic phase during winter [39]. Therefore, emotional cognitive biases in SAD during summer represent an important gap in research. Winter depression is the most common type of seasonality, with symptoms subsiding in summer [31]. However, it is noteworthy that the syndrome does not always follow this pattern. Some affected individuals experience incomplete summer remission [40] and could therefore display emotional cognitive bias in summer. Individuals with SAD, subsyndromal SAD, and individuals with incomplete summer remission of SAD might be suffering from significantly different clinical syndromes [40]. This heterogeneity was attempted to be explained in the context of the dual-vulnerability hypothesis [41]. This theory predicts that the risk for SAD is higher when people have seasonal physiological symptoms and at the same time a vulnerability to experiencing secondary depressive symptoms. The dual-vulnerability hypothesis builds on the idea that seasonality is a trait, where SAD is the extreme form of it [40,42]. According to the dual-vulnerability hypothesis, different sub-forms of SAD can be derived depending on the manifestation of the seasonality or depressive trait [40]. Consequently, cognitive emotional bias, as in depression, should be present only in a subgroup of affected individuals who score high on the depressive dimension of the syndrome.

In one study in individuals with SAD, reduction in recall of positive words from summer to winter predicted more severe depression symptoms in winter [34]. In this study, individuals with SAD differed from healthy controls by a significantly larger decrease in remembering positive words from summer to winter. However, according to descriptive statistics from that study, this effect might be partially explained by a relatively prominent increase from summer to winter in recall among controls, in addition to a slight decrease for SAD individuals. Additionally, in this study a slight decrease in memory performance was documented for all emotional categories in SAD individuals, and we speculate that this might also be attributable to a general memory impairment that is associated with depressive symptoms among SAD individuals in winter [31].

It should be noted that not all researchers have found negativity bias in winter among SAD participants [43], and usually no such bias is reported when investigating individuals with elevated seasonality symptoms in the time period when they feel usually good, i.e., during summer [39,44]. However, visual memory deficits were detected in the summer remission phase of SAD even in the absence of depressive symptoms [45]. Another study reported that individuals with SAD were significantly slower to color-name negative and season-related words in comparison to neutral words, both in winter and in summer [46].

There are also indications for anomalies in brain activity when comparing individuals with SAD to controls. Without alterations in behavioral measures, amygdala activation is altered in people with SAD both during the symptomatic and asymptomatic phase [47]. Individuals with elevated self-reported scores of seasonality show lower EEG power in a broad frequency range including delta, theta, and alpha activity [48]. Differences in brain activity can be found also in specific frequency bands during summer when comparing individuals with SAD to controls, but results have been conflicting. Specifically, while one research group reported increased power in delta, theta, and alpha bands [49], other research found lower power in the alpha frequency range [39]. Furthermore, differences in EEG activity during the remission phase indicate that neural responses to emotional stimuli differ between individuals with elevated risk for SAD and controls even in the absence of a behavioral effect [39]. Based on these findings, we hypothesize that an all-year-round altered disposition of individuals with SAD can be detected via physiological measures, but it is still unclear which cognitive processes are affected by this disposition.

Assessing physiological correlates of emotional responses with the electroencephalogram (EEG) has the advantage over functional magnetic resonance imaging of measuring brain responses to stimuli at a higher time-resolution, i.e., at a millisecond range [50]. This is important when a distinction between attention and memory effects is required, as attentional effects happen earlier, while memory effects such as recognition are detected later in brain correlates of cognitive processing. While the EEG has a disadvantage in spatial resolution as compared to functional magnetic resonance imaging, it can still detect the local effects of attention and memory. Specifically, measurements of alpha band power reactivity show a reduction in activity in relevant brain regions in response to tasks that demand attention [51]. The prefrontal lobe’s importance for emotional valence discrimination can be documented with the EEG [52]. Although reported time-windows for valence effects are conflicting [53], a commonly accepted finding is that event-related alpha oscillations in frontal and central regions tend to be enhanced during tasks involving memory [54] in a later time-window of 500–1000 ms [55]. As such, it was suggested that the largest valence effects during recognition can be measured after 400–800 ms [56]. In contrast, attention effects occur earlier, as reflected in alterations of alpha band power at 100–300 ms [57,58]. EEG alpha activity was even reported to reflect attentional effects as early as 100 ms post stimulus [57].

The EEG is also a commonly used biomarker in research on depression [59]. EEG research is much less established in SAD, but EEG biomarkers were successfully used in research on seasonality [60,61]. Lower power in most frequency bands as well as frontal asymmetrical activity was reported for participants with seasonality [48]. These findings once more emphasize the similarity between SAD and depression, since frontal alpha asymmetry was proposed as a trait marker for depression [62,63,64].

We hypothesize that in individuals who report elevated scores of seasonal symptoms, there is an all-year alternated emotional processing that might or might not be detectable in behavioral measurements, but we expect differences in responses of brain activity to emotional stimuli. Since it is still unclear whether cognitive processing vulnerabilities consist in emotional attention biases or emotional memory biases, in the present research we sought to answer the following research questions:Do people with high seasonality scores demonstrate an attentional and/or recognition bias towards negative valence compared to those with low seasonality scores during the remission phase in summer? We measured these behavioral correlates of emotional cognitive biases by means of reaction times during learning, and accuracy during recognition;In brain activity measured in summer, do people with high seasonality scores exhibit only differential effects during emotional attention, or is there an additional emotional effect during retrieval of memories? We measured frontal EEG alpha activity as a correlate for emotional processing and differentiated attention effects in an early (100–300 ms) time-window from memory effects in a late (400–800 ms) time-window during learning and recognition of emotional pictures. We expected that emotional attention bias at 100–300 ms should be measurable in both tasks, learning and recognition. In contrast, memory effects at 400–800 ms should emerge as significant differences between old and new stimuli in the recognition task.

By comparing groups who report high vs. low levels of seasonality during remission in summer we aim to determine whether there exist all-year-round EEG-differences in an early time-window, only, supporting a previously reported emotional attention bias, while additional differences in a later time-window linked to processes of memory retrieval (i.e., recognition) suggest that there might be additional emotional processing biases during retention and retrieval of emotional memory.

## 2. Materials and Methods

This study was part of a larger study, and part of the data was analyzed in a different context and published previously [39]. A flow diagram of the overall procedure is given inn Figure 1.

### 2.1. Recruitment

Participants were recruited by advertising on social media and among students. Requirement for participation was age of at least 18 years as well as fluency in Icelandic, as materials and instructions were in Icelandic. Participants were compensated with a voucher worth 4000 ISK.

### 2.2. Questionnaire

The Seasonal Pattern Assessment Questionnaire (SPAQ) was used to obtain the global seasonality score (GSS). Based on this score we estimated severity of seasonality among participants. The SPAQ is an 8-item self-administered questionnaire intended to evaluate seasonal changes in energy, mood, sociability, appetite, and sleep [65]. It was developed for the purpose of screening for seasonality and is not sufficient for a clinical diagnosis [66,67]. The questionnaire has overall good psychometric properties, including internal consistency, score distribution, factor structure, and test-retest reliability [66]. The Icelandic version we used in the present study portrays a positive predictive value for combined groups of SAD and subsyndromal SAD of 45%, a specificity of 73%, and sensitivity of 94% [68]. The scale ranges from 0–24, where a cut-off GSS to identify seasonality is ≥11 [69]. In this study, individuals with a GSS ≤ 10 were classified to be in a group of low seasonality and those with a GSS ≥ 11 in a group of high seasonality.

### 2.3. Picture Learning Condition

The picture learning condition was described in a previous publication [39]. In short, participants were shown 20 negative, 20 positive, and 20 neutral pictures (see Figure 2 for examples) from the open affective standardized image set, shortly “OASIS” [70] Stimuli were presented for at least 2000 ms while brain responses were measured. All participants saw the same list of pictures, but the list was randomized individually for each participant. The pictures were balanced by low, medium, and high arousal ratings. To maintain attention and enhance thinking about seasons, participants were asked to rate the season represented in each picture via button press on a standard computer keyboard (key V for spring, key B for summer, key N for fall, and key M for winter). Each trial ended with the participant’s button press, but at the earliest after 2000 ms. Thus, trials lasted until participants took a decision on the season represented in the picture. There was an inter-trial interval of 1000 ms plus a variance of 0–10 screen flip intervals (0–10 ms). For more details, see [39].

### 2.4. Recognition Task

The recognition task was administered as an old-new picture task. The 60 pictures learned in the previous condition were presented again but randomly intermixed with 60 new pictures that were not previously shown. Again, all participants received the same set of old and new stimuli, but the randomization of the order of presentation of pictures was individual. Thus, every participant saw the same set of old and the same set of new pictures. The same presentation timing was used as in the learning task. The 60 new pictures were balanced in emotional valence and arousal, paralleled with the 60 pictures from the learning session. Furthermore, according to the rating of four pilot participants, we balanced the two picture sets of old and new pictures by the season that those pilot participants rated the pictures for, to account for the attention-maintaining season-rating task during learning. In the recognition task, participants were instructed to press a key to indicate whether a picture was new (key: cursor to the right) or previously presented to them (i.e., old, key: cursor to the left). Each trial ended with the participant’s response and was followed by an inter-trial interval of 1000 ms plus a variance of 0–10 screen flip intervals (0–10 ms).

### 2.5. EEG Recording and Analysis

EEG was recorded with a passive 32-channel EEG system, the software Brain vision Recorder (Version 1.25, Brain Products GmbH, Gilching, Germany), and an Easy Cap for holding the electrodes in place. Passive Ag/AgCl electrodes (Fp1, Fp2, F3, F4, C3, C4, P3, P4, O1, O2, F7, F8, T7, T8, P7, P8, Fz, Cz, Pz, FC1, FC2, CP1, CP2, FC5, FC6, CP5, CP6, FT9, FT10, TP9, TP10, with a reference at FCz and Ground at AFz) were placed and recorded in an extended 10–20 system at a sampling rate of 1 kHz. The skin below the electrodes was prepared with an abrasive electrolyte, such that all impedances were below 10 kΩ. To measure the lower vertical electrooculogram, an additional electrode was placed below the right eye.

We analyzed EEG data with the software Brain Vision Analyzer (Version 2.0, Brain Products GmbH, Gilching, Germany). Firstly, EEG data were re-referenced against the common average reference, then filtered between 0.5 and 48 Hz (zero-phase shift Butterworth filters). Following that, we performed an ocular correction via Independent Component Analysis (ICA) and a back-transform with the infomax restricted algorithm. We excluded further artefacts according to the following criteria:check gradient: maximal allowed voltage step 50 µV/ms;check difference: maximal allowed difference of values in intervals of 200 ms: 200 µV;lowest activity allowed in 100 ms intervals: 0.5 µV.

We excluded ± 200 ms around the identified artefacts. Then, data were segmented from −500 ms before to 1000 ms after stimulus. These segments were band-pass filtered from 8–12 Hz and the resulting segments were averaged. From this average, alpha band activity in two time-windows was extracted, 100–300 ms and 400–800 ms. The measure per time-window was obtained as the mean activity over time in µV (unsigned values, rectified). Finally, we averaged EEG activity over frontal left electrodes (F3, F7) and frontal right electrodes (F4, F8).

### 2.6. Statistics

All statistical analysis was carried out with R/R-Studio [71].

First, we compared the groups of individuals with low seasonality and high seasonality in terms of age with a two-sample Wilcoxon test (Wilcoxon rank sum test or Mann–Whitney test). Distribution of gender between these two groups was compared using the chi-squared test.

We used 2-sample *t*-tests to compare reaction times for each valence category (negative, neutral, positive) and each season rating (spring, summer, fall, winter) between groups of low and high seasonality. Resulting *p*-values were corrected with Bonferroni correction (i.e., for the 7 tests, the critical *p*-value was *p* < 0.007). In addition, to control for effects introduced by the season-rating task, we counted how many positive, negative, and neutral pictures were rated to belong to spring, summer, fall, and winter. These numbers were analyzed with a semi-parametric repeated measures ANOVA that performs well under conditions of non-normality and heteroscedastic variances, implemented in the R package MANOVA.RM [72,73]. Because of unequal group sizes, we used a parametric bootstrap with 1000 repetitions, where the algorithm considers the group sizes during resampling. The grouping factor seasonality as well as the repeated-measures factors season and valence were included into this model.

Behavioral data during recognition were analyzed by calculating proportions of correctly recognized old or new pictures among all pictures for the three valence categories negative, neutral, and positive separately. Based on these proportions, we conducted the same semi-parametric repeated measures ANOVA with seasonality as a grouping factor and emotional valence as repeated-measures factor. Additionally, we compared the two seasonality groups with Mann–Whitney U tests for absolute numbers of each possible outcome of ratings for each valence category. This analysis was added because we expected that negative stimuli would lead to more mistakes in the sense of false memories (incorrectly rated as old), according to the literature. Thus, we conducted group comparisons for each valence category and the four possible outcomes: correctly rated as old, correctly rated as new, incorrectly rated as old, and incorrectly rated as new. Finally, we also conducted tests for the proportion of correctly rated images for each valence category. The resulting *p*-values were interpreted at the Bonferroni-corrected level of significance.

EEG data were analyzed with the same semi-parametric repeated-measures ANOVA as behavioral data (MANOVA.RM). We conducted the analysis separately for the learning and recognition condition. For learning, we analyzed only the early time-window (100–300 ms), to examine a potential attentional bias. For recognition, we included an additional repeated-measures factor time-window with the early (100–300 ms) and late time-window (400–800 ms) as factor levels, to differentiate between early attentional and late memory effects. Furthermore, for recognition we included the repeated-measures factor condition (old vs. new). In both analyses, we included the between-subject factor seasonality (low, high), and within-subject factors valence (negative, neutral, positive) and hemisphere (left, right).

Significant main effects and interactions of interest to our research questions were followed up with *t*-tests as post hoc tests. This was realized by averaging over the non-significant factors and contrasting the significant factor steps. All these contrasts/post-hoc tests were interpreted at the respective Bonferroni-corrected level of significance (e.g., for 6 post hoc tests, the corrected level of significance was *p* < 0.008, or for 18 post hoc tests it was *p* < 0.002).

Since the group with elevated seasonality scores was significantly younger than the group with low seasonality scores, we carried out a supplementary analysis to examine the effect of age (see Appendix A). To this end, we first grouped individuals into a younger group of up to 50 years and an older group of 50 years and older. This grouping is in line with a previous publication that showed that EEG responses in relation to seasonality interact with age [60]. According to this grouping, only three participants ended up in the group that scored high on seasonality and was older than 50 years. Since this group size is too small to allow for the age factor to be part of the statistical model, we performed the EEG analysis by excluding all participants who were older than 50 years (see Appendix A).

## 3. Results

### 3.1. Sample

Out of initially 119 participants, nine were excluded because the EEG data of the recognition task were missing (Nos. 30, 41, and 48), because the SPAQ was not completed, and thus, no grouping could be performed (Nos. 2 and 76), and because those individuals indicated feeling worst in summer (Nos. 8, 35, 36, and 94). The last exclusion criterion was important, since we performed examinations in the summer, assuming individuals who experience low mood in winter would feel comparatively good during the experiment.

There were N = 76 individuals with a GSS scores below 11 (median 4, range 0–10, hereafter “low-seasonality group”) and N = 34 individuals with a GSS score of 11 and higher (median 14, range 11–22, hereafter “high-seasonality group”). The low-seasonality group had a median age of 31 (range 18–66) and was therefore significantly older (W = 1365.5; *p* = 0.04) than the high-seasonality group with a median age of 27 (range 19–66). There were 61 women in the low-seasonality group and 27 women in the high-seasonality group, such that the difference in gender distribution was not significant (χ^2^ = 0.011; *p* = 0.918).

### 3.2. Behavioral Data

During learning, there were no significant differences in reaction time between groups for any valence picture category or any seasonality rating.

According to the semi-parametric ANOVA, no effect of seasonality (low vs. high) was found for the ratings of season perceived in the pictures, but significant effects for valence (F(2) = 223,277,051,948,506,112; *p* < 0.001), season (F(3) = 108.93; *p* < 0.001), and an interaction of valence and season (F(6) = 45.13; *p* < 0.001). Figure 3 illustrates the interaction and shows that there were only small differences in ratings for spring and winter, but positive pictures were rated less often as containing the season fall and more often as containing the season summer.

During recognition, there was no overall main effect for seasonality, but a significant main effect for valence (F(2) = 6.56; *p* = 0.002) which was such that the proportions of correctly recognized pictures was higher for neutral than for negative or positive stimuli (see Figure 4).

None of the single comparisons per valence category and memory outcome yielded a significant difference between the two groups in post-hoc Mann–Whitney U tests. While we expected to find more new negative pictures to be incorrectly rated as old, this was not the case. Only when taking together the correct responses for old and new pictures, a tendency was observed for a difference by seasonality group for the proportion of correctly recognized negative pictures among all negative pictures (W = 1025.5; *p* = 0.097). According to this tendency, there was a higher proportion of correctly recognized negative pictures in the high seasonality group as compared to the low seasonality group.

### 3.3. EEG Data

#### 3.3.1. Learning Condition

Table 1 shows the results of the semi-parametric repeated-measures ANOVA for the learning condition. None of the main effects or interactions were significant in the learning condition during the early (100–300 ms) time-window. The supplementary analysis that was limited to the sample up to 50 years of age led to the same result, with no significant effects or interactions.

#### 3.3.2. Recognition Condition

Table 2 shows the results of the semi-parametric repeated-measures ANOVA for the recognition condition.

The interaction between seasonality and valence during recognition consisted in a significantly higher EEG alpha power when viewing negative as compared to neutral pictures among individuals with elevated seasonality scores but no significant difference between the other picture categories (t(33) = 4.45; *p* < 0.001). In contrast, individuals with low seasonality scores showed the same difference between negative and neutral pictures (t(75) = 2.94; *p* < 0.001), but in addition, a significantly higher activity in response to positive as compared to neutral pictures (t(75) = 4.23; *p* < 0.001). The result is illustrated in Figure 5.

The most interesting effect during recognition which was relevant to our hypothesis was interaction between seasonality, valence, and time-window (see Figure 6).

Post hoc tests revealed that during the early (100–300 ms) time-window EEG alpha power measured in individuals with elevated seasonality scores was significantly higher for negative as compared to neutral pictures (t(33) = 4.38; *p* < 0.001). In contrast, among individuals with low seasonality scores, EEG alpha response to positive pictures was significantly higher as compared to neutral pictures (t(75) = 4.08; *p* < 0.001). In the late time-window, no differences were significant after correcting for multiple comparisons.

The significant main effect of emotional valence was such that highest alpha power was measured for positive pictures, followed by negative pictures, and lowest activity was measured for neutral pictures. However, only the differences between positive and neutral pictures (t(109) = 4.74; *p* < 0.001) and between negative and neutral pictures (t(109) = 4.85; *p* < 0.001) were significant according to post hoc paired *t*-tests.

Furthermore, time-window turned out as significant main effect, with frontal alpha power being higher in the 100–300 ms time-window as compared to the 400–800 ms time-window.

The interaction between valence and time-window was such that frontal alpha activity during presentation of negative pictures vs. neutral pictures differed in the early time-window (t(109) = 4.21; *p* < 0.001) but not in the late time-window, with lower alpha power for neutral pictures. Also, the difference between neutral and positive pictures differed significantly in the early time-window (t(109) = 4.36; *p* < 0.001), with lower alpha power for the neutral pictures.

Valence interacted significantly with condition (see Figure 7). We found that for old pictures none of the comparisons by valence were significant after correcting for multiple comparisons (*p* < 0.005). For new pictures, there was a significantly higher EEG alpha power response to negative as compared to neutral pictures (t(109) = 5.08; *p* < 0.001). EEG alpha power response was also significantly higher to positive new as compared to neutral new pictures (t(109) = 5.76; *p* < 0.001). In addition, when comparing old and new pictures, EEG alpha power was significantly higher for old neutral pictures as compared to new neutral pictures (t(109) = 2.97; *p* = 0.003).

The supplementary analysis, where we controlled for age by analyzing only the group that was up to 50 years old, yielded almost the same results. The main difference was that the interaction between seasonality and valence had a smaller effect size and, thus, was only by tendency significant (*p* = 0.072). In contrast, the interaction between valence, hemisphere, and time-window was stronger in the younger group, such that it reached significance (*p* = 0.023), while this was only a tendency in the analysis of the whole group.

## 4. Discussion

The intention of this research was to examine whether emotional cognitive bias was detectable for attentional and/or recognition memory processes in summer among people who are, according to a high global seasonality score, at risk to experience SAD during winter. According to reaction times, no attention bias was found in relation to elevated seasonality scores during learning, and no memory bias could be confirmed, either, although by tendency recognition performance was better for negative pictures in the high seasonality group. Frontal alpha power, however, showed a significant interaction with seasonality, which was apparent only during the recognition task but not during leaning. In the recognition task, the effect emerged in the early time-window, but not in the late time-window. Thus, in sum, EEG data suggest that there is a significant difference in emotional processing in individuals with elevated seasonality codes also during summer. However, our results remain inconclusive as to whether this is rather a bias in attention or memory.

### 4.1. Behavioral Results: Emotional Biases Independent of Seasonality

There was no clear effect in behavior when individuals with low and high seasonality scores were compared in summer. Some effects that were independent of seasonality are noteworthy. We found that pictures were more likely rated to reflect a scenario from summer when they were positive, while positive pictures were less likely ascribed to a scenario in fall. Since there was no interaction with seasonality, we cannot draw any conclusion whether this is due to fall being a reference for negative changes such as less nature contact [74] or rather related to a SAD-like drop in overall wellbeing and mood [31]. This result might be related to the stimulus material used, the OASIS database [70]. It is possible that the database includes more negative pictures that are more likely to be rated to represent the season fall. Future studies on the OASIS database should investigate the picture properties in more detail with respect to the semantic content of the pictures.

This study investigated the relation between seasonality and correct responses, as well as mistakes in recognition of emotional stimuli. Previous research has shown that patients with depression are more likely to rate stimuli of negative valence by mistake as old [36,37]. The present research did not replicate this finding among individuals who are at risk of experiencing seasonal affective disorder. Only the proportion of correctly recognized negative pictures differed by tendency between the groups, with a higher level of correctly remembered negative pictures in the group who reported higher levels of seasonal symptoms. Thus, we could not translate previous findings on false negative memories from depressive populations to individuals with elevated seasonality. We may speculate that there is a tendency for enhanced processing of negative stimuli, leading to better recognition performance in individuals with elevated seasonality scores. This is in contrast with previous research that reported lower overall recognition memory in individuals with winter depression during the symptomatic phase [30]. This divergence in results might be explained by at least two differences between the present work and prior studies. Firstly, prior research [30,33] was based on data that were collected during winter, as opposed to summer in our study. Secondly, also the stimulus material differed. While we used emotional pictures, earlier research implemented a recognition memory task for stories [30]. Another study in seasonal depression reported reduced recognition of happy facial expressions but increased recognition for negative personality characteristics [33], which is more in line with the tendential findings of our study.

### 4.2. Seasonality Effects in EEG Responses—Attention or Memory?

Several interesting effects occurred during recognition of emotional pictures. With respect to our hypothesis, it is most noteworthy that, only during the early (100–300 ms) time-window, brain responses to emotional stimuli differed between individuals with elevated seasonality from those with low seasonality scores. Specifically, people with elevated seasonality scores showed a significantly higher alpha power in response to negative as compared to neutral pictures, but no such difference for positive pictures. In contrast, for people with low seasonality scores, there was a significantly higher alpha power in response to positive as compared to neutral pictures, but no such difference for negative pictures. Since this difference appeared in the early time-window, our results could be interpreted as a bias in attention rather than memory for individuals with high seasonality during the remission phase in summer. However, if the reason for this interaction between seasonality, time-window, and valence was only that there is an effect of attention in the absence of a memory effect, we would have expected to find the same effect in the EEG data recorded during the learning task. In contrast, no effects were found during learning. There are several possible explanations for this unexpected result. Firstly, it is possible that the list of pictures that we used for learning, and which represented the old pictures in the recognition task, was less likely to exhibit valence effects as compared to the list of new pictures. Indeed, according to a condition x valence interaction, we found stronger differences for new as compared to old pictures. Although the picture lists were well balanced in terms of valence and arousal, there might be other unknown factors that we did not control for and that played a role. Secondly, attentional processes during learning might be entirely different from those during recognition. The task during learning was to rate the season of the pictures—we asked the participants to focus on seasonal cues, which was not always easy as not all pictures are obviously related to a season. During recognition, the simple old-new task allowed the participants to focus on familiarity of the pictures. These different foci of attention might have impacted the result. Thirdly, we could speculate that attention during encoding might be differentially affected by valence and seasonality as compared to attention during retrieval.

In previous research on SAD, recognition of emotional personality characteristics [33] and recognition of arousal stories [30] have been investigated. Old/new recognition tasks are commonly used in research on emotional recognition in relation to depression [36,75,76], but research on emotional recognition memory bias is rather scarce in relation to SAD.

Attentional bias in SAD was documented previously [30,33] in the form of an increased vigilance to negative stimuli, but only during winter. A relative decrease in alpha power signals neural activity which is linked to cognitive processes such as retrieval of semantic knowledge in the long-term memory and to good performance [77,78]. In our study, we did not measure a relative decrease in alpha power (i.e., event-related desynchronization, ERD), thus, comparison with studies investigating ERD are not directly possible. However, the difference in alpha activity between emotional vs. neutral pictures especially in the early (100–300 ms) time-window is in line with previously reported valence effects with an early, attention-relevant timing (<300 ms). Specifically, emotional valence effects can be found in early event-related potentials such as the P1 [79]. Moreover, valence discrimination has been reported to occur early (<300 ms) in EEG studies investigating event-related desynchronization/synchronization [80,81,82].

Previous studies on attentional bias in relation to depression have been contradictory [83]. It was theorized that attentional bias is only linked to anxiety, whereas memory bias is linked to depression or comorbidity of these disorders [84]. However, a meta-analysis revealed that overall there is an attentional bias for negative stimuli in depressed samples [83].

In sum, we cannot clearly answer the question whether there is an attentional and/or memory bias in processing of emotional stimuli in individuals with elevated seasonality scores in summer. Nevertheless, the fact that the differences appear rather early in stimulus processing suggests that attentional differences play an important role.

### 4.3. Emotional Bias in Summer for People with Elevated Seasonality Scores

It was reported that remitted patients with depression no longer exhibit a negative attentional bias but pay less attention to positive stimuli than controls [85]. This is in line with our findings. We found that, overall, individuals with low seasonality scores show elevated EEG alpha activity in response to positive and negative pictures as compared to neutral pictures. Thus, this group exhibited the expected emotional bias, with higher brain activity in response to emotional as compared to neutral pictures. In contrast, among individuals with elevated seasonality scores, the EEG alpha activity in response to positive pictures is overall as low as for neutral pictures, such that only the difference between negative and neutral pictures becomes significant. Our data, thus, support the theory that there is an all-year-round vulnerability of individuals who are at risk for seasonal affective disorder.

### 4.4. Physiological Responses in EEG Alpha Power

We found significantly higher frontal alpha power for emotional as compared to neutral pictures, but this was significant in the early time-window (100–300 ms), only. Additionally, EEG alpha power was overall higher in the early (100–300 ms) as compared to the late (400–800 ms) time-window. Firstly, reactivity in the alpha range is plausibly a correlate of emotional processing, as other research reported higher frontal alpha power in response to positive and happy emotions as compared to negative emotions [86,87]. In addition, alpha power may also vary across negative emotions such as anger vs. fear and sorrow [88]. Secondly, these results are in line with previous research, showing larger alpha activity in response to emotional stimuli within an early time frame [89]. We may speculate that this result is due to early top-down attention mechanisms which are reflected as increased alpha activity that actively inhibits irrelevant information to prioritize relevant information [90].

Furthermore, frontal alpha power was overall higher in the early (100–300 ms) time-window as compared to the late time-window, which is easily explained by the alpha desynchronization that happens typically at a delay of 500–600 ms after stimulus presentation [91,92,93].

In contrast to previous research that revealed asymmetric EEG alpha responses in relation to emotional processing [94], we did not find an interaction between hemisphere and valence or between hemisphere and seasonality. Preceding studies have shown robust evidence suggesting different hemispheric roles in emotional processing and regulation [95], where the left hemisphere is linked to positive affect and the right hemisphere to negative affect [96]. Among individuals with depression, frontal lobe asymmetry is common, with lower left activation in relation to withdrawal from aversive stressors [97]. Frontal hemispheric asymmetries in the alpha range can also be detected during valence discrimination, with increased left frontal responsivity to positive stimuli and right frontal responsivity to negative stimuli at about 600–1200 ms post-stimulus [81]. Similarly, left hemispheric elevated event-related synchronization in the time-window of 800–1000 ms in the alpha frequency range is induced by negative stimuli but not neutral or positive stimuli [56]. It is, thus, possible that the time-windows examined in the present study were overall too early to detect hemispheric differences. Additionally, our supplementary analysis points to a significant interaction between valence, hemisphere, and time-window when restricting the sample to the younger participants. We speculate that hemispheric effects might be more present in the younger group as compared to the older group of participants.

### 4.5. Limitations

One limitation we discussed earlier is the potential bias in the list of old and new pictures. Counterbalancing the old and new picture lists across participants could have prevented the problem. Future experiments should take this issue into account.

Young people and evening chronotypes are especially susceptible to experiencing winter depression [98], and there is evidence for differential effects of seasonality on EEG responses to mood induction [60]. In our sample, individuals with increased levels of seasonality were significantly younger than individuals with lower levels of seasonality. In the supplementary analysis, we found that the results were overall the same, but with a stronger hemispheric difference. Further studies should aim at recruiting more individuals at advanced age with elevated seasonality scores to examine this potential confounder in more detail.

In this study, we did not perform a clinical diagnosis of SAD but used a screening tool to identify people who are—by self-report—at risk of suffering from SAD during winter. The actual number of people experiencing SAD in this group in the consecutive winter is likely to be lower than those who reported themselves to be at risk. Therefore, our results cannot be directly compared to other research that examined individuals with SAD during remission.

## 5. Conclusions

Although there was no indication in our data that individuals who were at risk of experiencing SAD during winter would exhibit biased memory performance in summer, according to frontal EEG alpha power there is an indication for early attentional processing differences for emotional stimuli. Since this effect was detectable only in the recognition task, but not in the learning task, we cannot clearly conclude that attentional biases contributed to an all-year-round vulnerability of individuals with SAD. However, we could confirm that there is altered processing of emotional stimuli in SAD already during summer, in line with what would be expected from depressed individuals. According to our data, these differences are likely to be found in an early, attention-relevant time-window in the EEG alpha band. The present study opens the stage for future clinical trials to investigate the effectiveness of cognitive-behavioral interventions that target emotional biases already in summer to prevent winter depression among those who are at risk.

## Figures and Tables

**Figure 1 brainsci-14-00002-f001:**
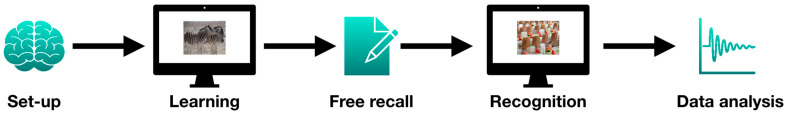
Flow diagram of the overall procedure.

**Figure 2 brainsci-14-00002-f002:**
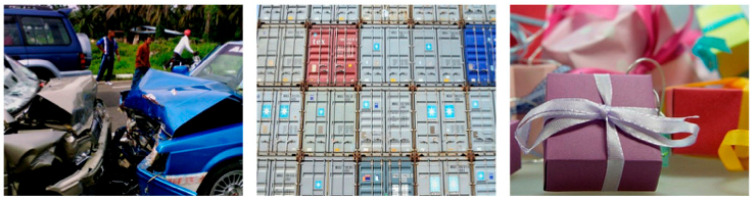
Examples for negative (**left**), positive (**right**), and neutral (**center**) pictures from the open affective standardized image set [71].

**Figure 3 brainsci-14-00002-f003:**
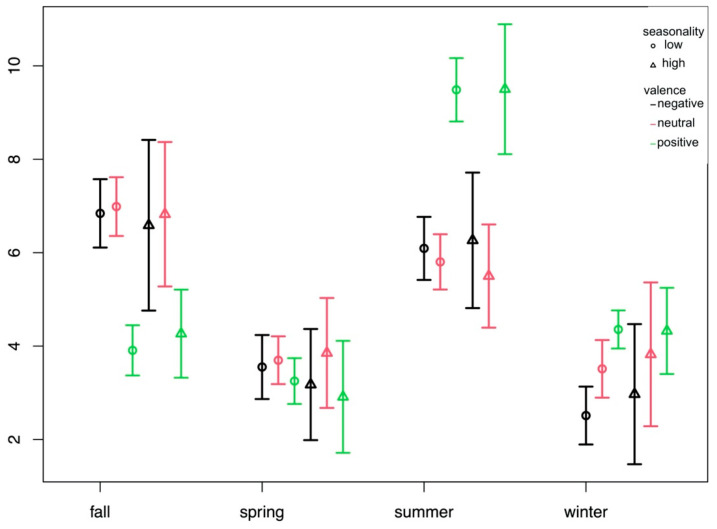
Average number (*y*-axis) of ratings of season (*x*-axis) for pictures of negative, neutral, and positive valence, separately for groups with low (<11) and high (>10) seasonality scores according to the seasonal pattern assessment questionnaire (SPAQ).

**Figure 4 brainsci-14-00002-f004:**
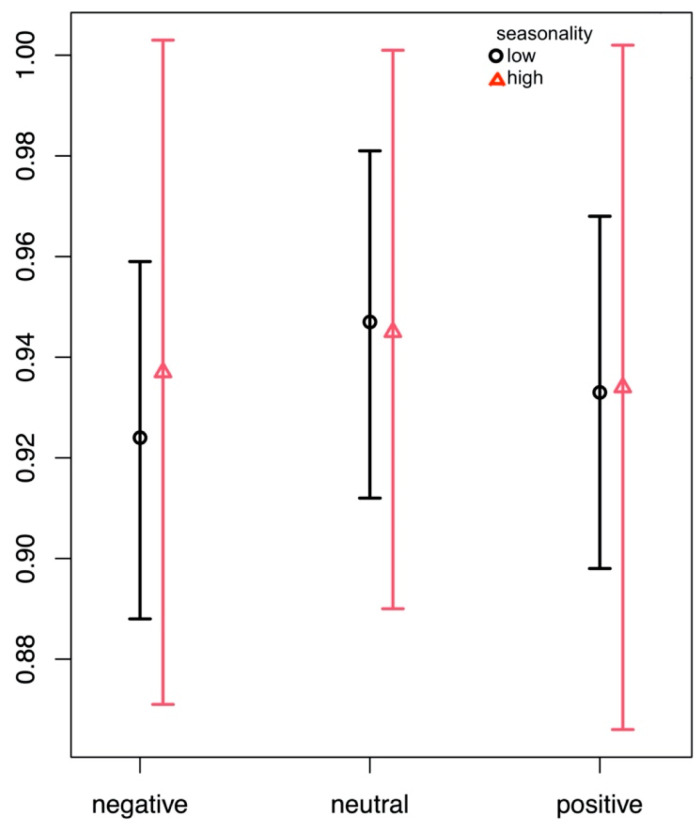
Proportions of correctly recognized pictures (*y*-axis) of negative, neutral, and positive valence, separately for groups with low (<10) and high (>9) seasonality scores according to the seasonal pattern assessment questionnaire (SPAQ).

**Figure 5 brainsci-14-00002-f005:**
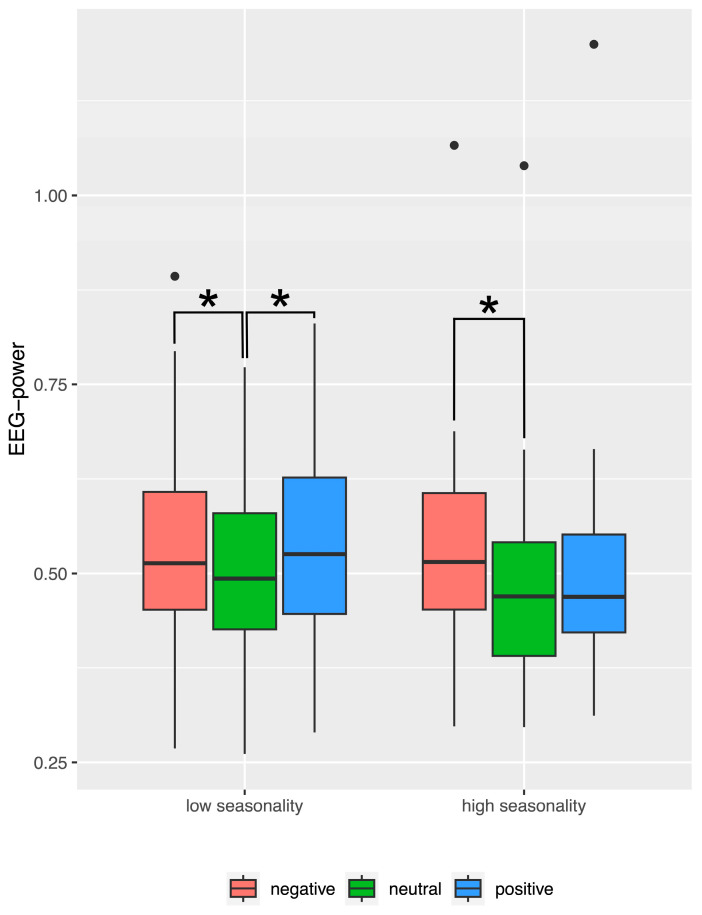
Boxplots for frontal EEG alpha power during recognition, averaged across hemispheres (left and right) and time-windows (100–300 ms, 400–800 ms), illustrating the significant interaction between picture valence and seasonality group. Asterisks indicate significant post hoc tests. Lower and upper hinges correspond to the first and third quartiles (25th and 75th percentiles). The whiskers extend from the hinge to the largest/smallest value but no further than 1.5 times the interquartile range from the hinge. Data beyond the end of the whiskers are plotted as individual outliers (black dots).

**Figure 6 brainsci-14-00002-f006:**
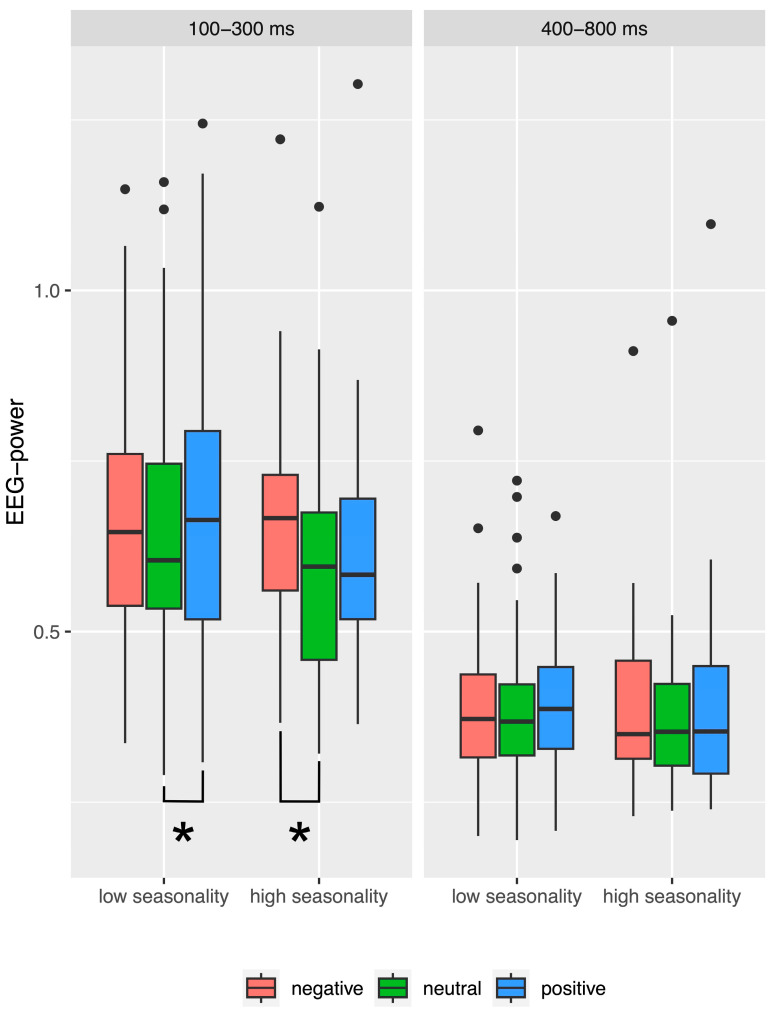
Boxplots for frontal EEG alpha power during recognition, averaged across hemispheres (left and right) separately for time-windows (100–300 ms, 400–800 ms), illustrating the significant interaction between picture valence (negative, neutral, and positive pictures), seasonality group (low and high seasonality), and time-window. Asterisks indicate significant post hoc tests. Lower and upper hinges correspond to the first and third quartiles (25th and 75th percentiles). The whiskers extend from the hinge to the largest/smallest value but no further than 1.5 times the interquartile range from the hinge. Data beyond the end of the whiskers are plotted as individual outliers (black dots).

**Figure 7 brainsci-14-00002-f007:**
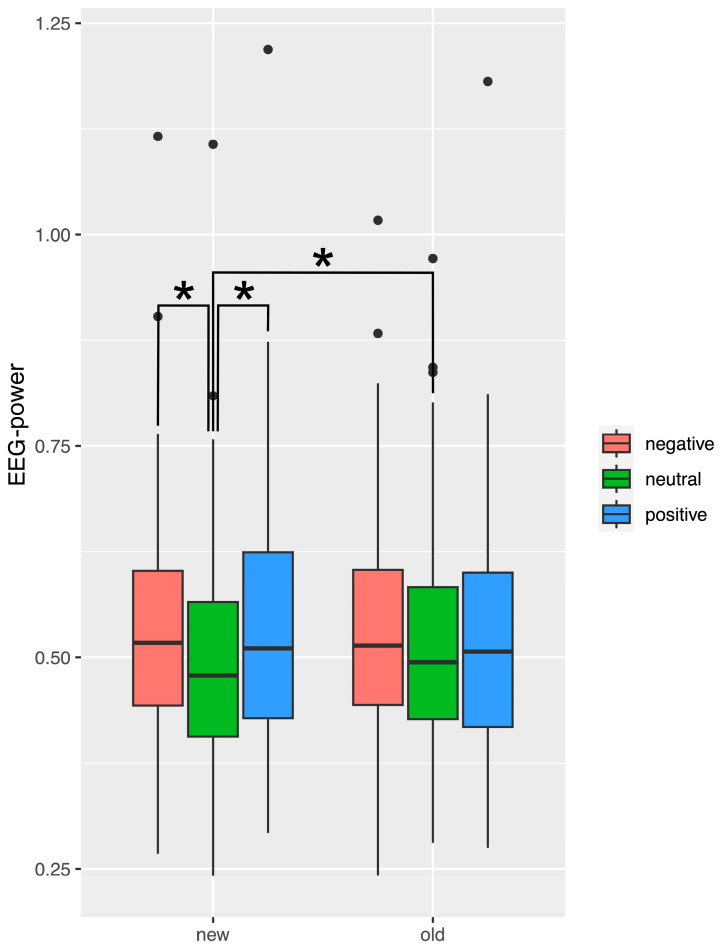
Boxplots for frontal EEG alpha power, averaged across hemispheres (left and right) and time-window (100–300 ms, 400–800 ms), illustrating the significant interaction between picture valence and condition in the recognition task (old vs. new pictures). Lower and upper hinges correspond to the first and third quartiles (25th and 75th percentiles). Asterisks indicate significant post hoc tests. The whiskers extend from the hinge to the largest/smallest value but no further than 1.5 times the interquartile range from the hinge. Data beyond the end of the whiskers are plotted as individual outliers (black dots).

**Table 1 brainsci-14-00002-t001:** Results of the semi-parametric repeated-measures ANOVA for learning, with between-subject factor seasonality (low, high), and within-subject factors valence (negative, neutral, positive) and hemisphere (left, right).

Factor or Interaction	*F*	*df*	*p*	*res.p* ^1^
seasonality	1.20	1, 163.15	0.274	0.304
valence	1.07	1.99, Inf	0.343	0.359
hemisphere	1.73	1, Inf	0.188	0.198
seasonality × valence	1.78	1.99, Inf	0.169	0.185
seasonality × hemisphere	1.48	1, Inf	0.224	0.217
valence × hemisphere	1.53	1.98, Inf	0.218	0.222
seasonality × valence × hemisphere	0.21	1.98, Inf	0.806	0.791

^1^ Resampling *p*-value obtained with parametric resampling and 1000 repetitions.

**Table 2 brainsci-14-00002-t002:** Results of the semi-parametric repeated-measures ANOVA for recognition, with between-subject factor seasonality (low, high), and within-subject factors valence (negative, neutral, positive), hemisphere (left, right), time-window (100–300 ms, 400–800 ms), and condition (old, new).

Factor or Interaction	*F*	*df*	*p*	*res.p* ^1^
seasonality	0.55	1, 150.71	0.458	0.477
valence	15.76	1.96, Inf	<0.001	<0.001
hemisphere	1.03	1, Inf	0.309	0.336
time-window	460.01	1, Inf	<0.001	<0.001
condition	0.30	1, Inf	0.585	0.589
seasonality × valence	3.64	1.96, Inf	0.027	0.022
seasonality × hemisphere	0.18	1, Inf	0.671	0.672
seasonality × time-window	3.50	1, Inf	0.061	0.082
seasonality × condition	0.95	1, Inf	0.329	0.349
valence × hemisphere	0.51	1.98, Inf	0.596	0.576
valence × time-window	5.42	1.94, Inf	0.005	0.002
valence × condition	6.93	1.89, Inf	0.001	<0.001
hemisphere × time-window	0.22	1, Inf	0.638	0.612
hemisphere × condition	0.33	1, Inf	0.566	0.564
time-window × condition	1.25	1, Inf	0.264	0.277
seasonality × valence × hemisphere	0.35	1.98, Inf	0.706	0.721
seasonality × valence × time-window	3.84	1.94, Inf	0.023	0.023
seasonality × valence × condition	0.13	1.89, Inf	0.865	0.878
seasonality × hemisphere × time-window	0.03	1, Inf	0.862	0.884
seasonality × hemisphere × condition	0.10	1, Inf	0.749	0.758
seasonality × time-window × condition	0.25	1, Inf	0.620	0.621
valence × hemisphere × time-window	2.81	1.97, Inf	0.061	0.070
valence × hemisphere × condition	0.18	1.9, Inf	0.828	0.823
valence × time-window × condition	1.04	1.93, Inf	0.350	0.363
hemisphere × time-window × condition	0.12	1, Inf	0.726	0.720
seasonality × valence × hemisphere × time-window	0.09	1.97, Inf	0.914	0.912
seasonality × valence × hemisphere × condition	0.44	1.9, Inf	0.635	0.661
seasonality × valence × time-window × condition	0.42	1.93, Inf	0.652	0.648
seasonality × hemisphere × time-window × condition	0.04	1, Inf	0.842	0.824
valence × hemisphere × time-window × condition	0.23	1.97, Inf	0.789	0.796
seasonality × valence × hemisphere × time-window × condition	0.12	1.97, Inf	0.882	0.883

^1^ Resampling *p*-value obtained with parametric resampling and 1000 repetitions.

## Data Availability

The data generated for this study are available in the Appendix A (S2-data.zip).

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
