# Peer review of "Emotional Bias among Individuals at Risk for Seasonal Affective Disorder—An EEG Study during Remission in Summer"

_brainsci, 2023, doi:10.3390/brainsci14010002_

Round 1

Reviewer 1 Report

Comments and Suggestions for Authors Manuscript ID: brainsci-2756414   Manuscript: "Emotional bias among individuals at risk for seasonal affective disorder—an EEG study"  

Authors: Theodorsdottir, D. & Holler, Y.

In the reviewed article, the authors examine the effect of seasonal affective disorder on emotional biases while in remission (i.e., summer months). This is a very interesting research question and one in which the use of neural measures may be particularly important. However, I had several concerns with the analysis approach that the authors used to identify memory effects in both the behavioral and neural data. I also felt that the authors could have provided more background information about the effects that they were studying. I discuss these and other concerns below:

1.     The authors do not appear to have theoretically-driven hypotheses regarding the neural patterns that they might see in the EEG data. Instead, the hypotheses appear to be focused on whether unspecified effects will be seen in the “attention” or “memory” windows. However, the introduction does not explain why such a distinction might be important. The authors state that “it is still unclear which cognitive processes are affected by this disposition”, but not how these processes differ, why that might matter for depression, or how other studies have failed to study it. It might help to have a paragraph dedicated to each process that includes prior research on emotional bias and depression effects.

2.     Related to the point above, it would be helpful if the authors spent more time elaborating on the prior work showing memory differences during summer (i.e., refs 32, 33, 34, and 28). Currently, these studies are embedded in a paragraph focusing on the lack of research in summer, but it seems like a good bit is already known about the SAD remission phase. Please be more specific about how the current research builds on these ideas. In particular, looking at the references, it appears that reference 28 might actually be a paper from this same dataset. Is this the case? If so, that should be made explicit in the manuscript.

3.     In addition to not setting up the attention v. memory question in the introduction, the authors do not set themselves up well to identify memory effects in either the behavioral or neural data. The analyses are not the standard analyses used in the memory literature and may be interfering with their ability to capture real differences. If the authors have a reason to use these atypical analyses, this reason should be explained in the manuscript. If not, I would recommend that they analyze the data in a more traditional way. For instance:

a.     For the behavioral memory data, the authors appear to be averaging together hits and correct rejections to create a composite score of accurate responses. This introduces two problems in the analysis. First, the scores here are quite high and may be subject to a ceiling effect, making it impossible to see real differences across conditions. Traditionally, memory accuracy is calculated by subtracting false alarm rates from hit rates. Doing so will bring your dependent variable down so that they won’t be as likely to be affected by ceiling effects. Second, averaging the two together means that the authors are unable to look at false alarm rates independently. Both the introduction and discussion sections highlight differences in negative false alarms in depression, so this might be an important measure.

b.     I had several concerns about the EEG data:

                                               i.     First, it was unclear why the same time windows were being used for encoding processes and recognition processes. I have not seen anything in the past to suggest that the timescale for encoding and retrieval are the same and should be considered in parallel. The authors should look at the literature and establish whether this approach is appropriate and then clearly address this in the introduction.

                                              ii.     I was also surprised to see learning, old recognition, and new recognition included in the model as levels of the same factor. Participants are not completing the same task in learning and recognition, so comparing old v. new recognition is very different from comparing either of them to learning. Typically in EEG/ERP research on recognition, the only way to identify memory effects is to compare old v. new trials, so it might be stronger to first conduct an analysis with just old v. new recognition.

4.     In several key places (e.g., the abstract and the first paragraph of the discussion) the primary EEG result (i.e., the time by valence by group interaction) is not fully interpreted. Specifically, the valence component is not discussed and is unclear why the finding is important to understanding SAD

5.     There were some aspects of the methodology that were unclear:

a.     Throughout the abstract, introduction, and most of the methods, it was unclear in which phases of memory (encoding and/or retrieval) EEG data were being collected.

b.     The methods say that pictures were presented for “at least 2000ms” but it isn’t clear what the range or maximum presentation would be.

c.     The ITI included “a variance of 0-10 screen flip intervals” – what does this mean for the duration?

d.     Were new and old lists counterbalanced across participants? Or did all participants see the same lists of old and new items?

6.     The authors noted confounds of seasonality ratings (in the stimulus set) and age (in the group variable) but did not attempt to control for either in the analysis. It would be helpful to have supplementary analyses looking at how these confounds may contribute to the overall pattern of results.

More minor concerns:

7.     Please present the results of all EEG contrasts, not just the significant ones.

8.     In the figure captions, please note whether error bars are SE or CI.

Comments on the Quality of English Language

I would recommend the authors go through the manuscript with a goal of improving readability. Specifically, there are several long, compound sentences that could be trimmed to make it easier for the reader to follow along with key concepts. In addition, there are some sentences that are difficult to parse, grammatically.

For instance, on page 2: “While winter depression is the most common type of seasonality with symptoms subsiding in summer it is noteworthy that there is the syndrome does not always follow this pattern as some affected individuals experience incomplete summer remission “

Author Response

We enclose a PDF document with our responses to the reviewer.

Reviewer 2 Report

Comments and Suggestions for Authors

Title is not clear and it does not reflect the work. 

Abstract is too vague, it does not present the novelty of the proposed work.

Literature review is too short, authors must include critical analysis and present the research gaps.

Flow diagram of the proposed method is missing. It should be included for better readability. 

Authors should specify the dataset that they have used in this study. Detailed description of dataset must be included. 

Conclusion section is too short, authors should specify the contribution of the proposed method in the body of knowledge. 

Future directions are missing, authors must include the future work in the conclusion section. 

In the references, authors must include the recent literature. 

Comments on the Quality of English Language

English language proofreading is recommended. 

Author Response

We enclose a PDF document with the responses to the reviewer.

Round 2

Reviewer 2 Report

Comments and Suggestions for Authors

Minor English grammar checking is recommended. 

Comments on the Quality of English Language

Author have addressed most of my comments.